# Multitask Learning Based Intra-Mode Decision Framework for Versatile Video Coding

Naima Zouidi [1,2,*], Amina Kessentini [2,3,†], Wassim Hamidouche [1,†], Nouri Masmoudi [2,†] and Daniel Menard [1,†]

1    The Institute of Electronics and Digital Technologies, UMR CNRS 6164, Electronics and Industrial Computing Department, The National Institute of Applied Sciences, University of Rennes, 35000 Rennes, France

2    Laboratory of Electronics and Information Technologies, Electronic Department, National School of Engineers, University of Sfax, Sfax 3038, Tunisia

3    Electronics and Industrial Computing Department, Higher Institute of Computer Sciences and Multimedia, University of Gabes, Gabes 6033, Tunisia

*    Correspondence: naima.zouidi@insa-rennes.fr; Tel.:+33-0779866951

†    These authors contributed equally to this work.

**Abstract:** In mid-2020, the new international video coding standard, namely versatile video coding (VVC), was officially released by the Joint Video Expert Team (JVET). As its name indicates, the VVC enables a higher level of versatility with better compression performance compared to its predecessor, high-efficiency video coding (HEVC). VVC introduces several new coding tools like multiple reference lines (MRL) and matrix-weighted intra-prediction (MIP), along with several improvements on the block-based hybrid video coding scheme such as quatree with nested multi-type tree (QTMT) and finer-granularity intra-prediction modes (IPMs). Because finding the best encoding decisions is usually preceded by optimizing the rate distortion (RD) cost, introducing new coding tools or enhancing existing ones requires additional computations. In fact, the VVC is 31 times more complex than the HEVC. Therefore, this paper aims to reduce the computational complexity of the VVC. It establishes a large database for intra-prediction and proposes a multitask learning (MTL)-based intra-mode decision framework. Experimental results show that our proposal enables up to 30% of complexity reduction while slightly increasing the Bjontegaard bit rate (BD-BR).

**Keywords:** versatile video coding; intra-prediction; rate distortion; fast intra-prediction decision; multitask learning

## 1. Introduction

It is undeniable that video technology is moving into a new era. From the streaming of digitized content to the use of sophisticated Augmented Reality (AR), the video industry is evolving quickly [1]. In fact, many key concepts have been introduced into the market, such as overlaying of digital content in the life environment, shooting, sharing, and streaming 360° videos, and, certainly, the inclusive access to streaming platforms. Amid these revolutionary changes, video traffic over the Internet has quadrupled in only a few years, reaching roughly 82% of global IP traffic [1]. This rapid growth in video demand has made it crucial for organizations, such as International Telecommunications Union (ITU) and ISO/IEC Motion Picture Experts Group (MPEG), to urgently tackle the potential need for a more efficient video coding standard than High Efficiency Video Coding (HEVC). Thus, in July 2020, the new video coding standard, namely Versatile Video Coding (VVC), was released by the Joint Video Experts Team (JVET). Typically, versatile stands for various coding tools, that allow VVC to deliver high-quality videos at low bit rate cost and support a wide variety of media services. In fact, the VVC is able to encode Ultra-High Definition (UHD) and immersive video contents at nearly 40% of the bit rate saving compared to its predecessor, HEVC [2]. This outstanding compression performance, as mentioned, is essentially based on several improvements of the block-based hybrid

video coding scheme. The new Quadtree with Nested Multi-type Tree (QTMT) structure [3], for example, supports wide homogeneous regions in high spatial resolutions as well as rectangular narrow texture blocks. As for intra-prediction, 65 finer-granularity angular Intra Prediction Modes (IPMs) [3] with DC and planar features were introduced alongside several novelty coding tools. These latter tools include Multi-Reference Lines (MRL), low-complexity neural network-based intra-prediction features, also known as Matrix weighted Intra Prediction (MIP) and Intra Sub-Partitions (ISP). In addition, VVC supports new motion compensation techniques, such as affine motion compensation [4], from different control points. It also enhances transformation, quantization, and entropy coding with several tools, such as Low-Frequency Non-Separable Transform (LFNST), Multiple Transform Selection (MTS), dependent scalar quantization, and Context Adaptive Binary Arithmetic Coding (CABAC) [5].

Despite achieving substantial coding efficiency and wide coding support, the computational complexity remains a key challenge, especially when looking toward real-time implementation of the VVC codec on streaming or embedded devices. According to [2], the VVC Test Model (VTM), in All Intra (AI) configuration, is 31 times more complex than the HEVC Test Model (HM), which is technically beyond most streaming or embedded device capabilities. Under these circumstances, several works, such as [6,7], proposed a fast encoding decision algorithm to alleviate the computational complexity of the VVC. Although these works have helped reduce the computational complexity of the VVC, they are still unable to deal effectively with the high diversity of block shapes in the QTMT partitioning structure and the different intra-coding tools. To this end, this paper proposes a multitask learning-based intra-mode decision framework for VVC. It establish a large database for the new intra-coding tools and train a multitask learning Convolutional Neural Network (CNN) to reduce the number of tested IPMs based on the inferred top-2 intra-coding tools. Indeed, this framework has many advantages. It deals effectively with the high diversity of block shapes and can open up additional hardware optimizations, such as parallelism and GPU acceleration. Furthermore, our intra-mode decision framework simultaneously predicts the probability vectors for all the intra-coding tools, which reduces the model inference time.

The remainder of this paper is organized as follow. Section 2 highlights the key intra-coding tools of VVC, while reviewing the different steps of the intra-mode decision. Section 3 gives an overview of related works on fast encoding decision. Then, the proposed Multi-task Learning (MTL) based intra-mode decision framework is presented in Section 4. Finally, Section 5 concludes this paper.

## 2. Intra-Prediction

Intra-prediction is the step of video coding that deals with spatial redundancy [8]. It encountered several enhancements in VVC, such as the introduction of finer granularity IPMs. In this section, the key intra-coding tools and the intra-mode decision steps are reviewed.

### 2.1. New Intra-Coding Tools

In contrast to HEVC, which defines 33 angular IPMs, finer granularity intra-prediction is proposed in VVC to accommodate the directional structures more efficiently. Hence, angular IPMs were extended up to 65. However, DC and planar features remain in use. Figure 1 illustrates the new IPMs with green dotted arrows.

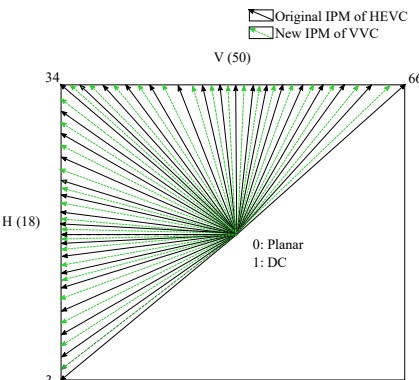

**Figure 1.** Illustration of the 67 IPMs. V and H are vertical and horizontal modes, respectively.

In addition to extending the number of angular IPMs, intra-prediction from nonadjacent reference samples, so-called Multi-Reference Lines (MRL) can be used to exploit farther regions in predicting sharp contents [3]. As shown in Figure 2, the nearest reference line is denoted with reference line 0 and farther reference lines with reference line 1 or 3. For instance, the used IPM, for MRL, is restricted to the list of nonplanar Most Probable Modes (MPMs) [3]. Moreover, Intra Sub-Partitions (ISP) can be used to adapt the intra-prediction to narrow texture structures by dividing the intra-predicted block horizontally or vertically into 2 or 4 sub-blocks, which share the same IPM [5].

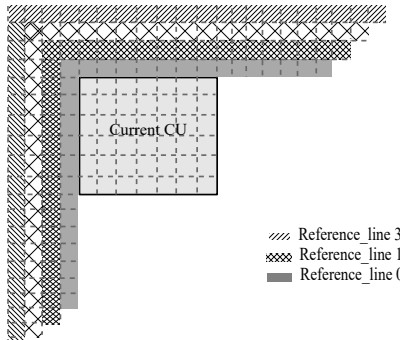

**Figure 2.** Reference lines neighbouring to an intra-predicted block.

In typical intra-prediction, samples are interpolated from reference samples according to the prediction direction [9]. However, the new Matrix weighted Intra Prediction (MIP) can use downsampled reference samples to generate a reduced prediction signal throughout a matrix-vector multiplication. Then, it interpolates the remaining samples by using the left and above reference samples to form the final prediction [5]. Indeed, the used matrix and offset vector can be selected from a set of 32 pretrained matrices and vectors, denoted as MIP modes.

### 2.2. IntraModeDecision

Similar to HEVC, the Rate-Distortion Optimization (RDO) of the cost $J$, computed in Equation (1) is used to infer the best IPM. For instance, the RDO reports for each encoding decision the tradeoff between distortion $D$ and rate (number of required bits) $R$, where $\lambda$ is the Lagrangian multiplier.

$$J = D + \lambda R \tag{1}$$

Because VVC enhances intra-prediction with several intra-coding tools, the intra-mode decision was also extended to handle the new intra-coding tools decision. Therefore, three main steps are required, as shown in Figure 3.

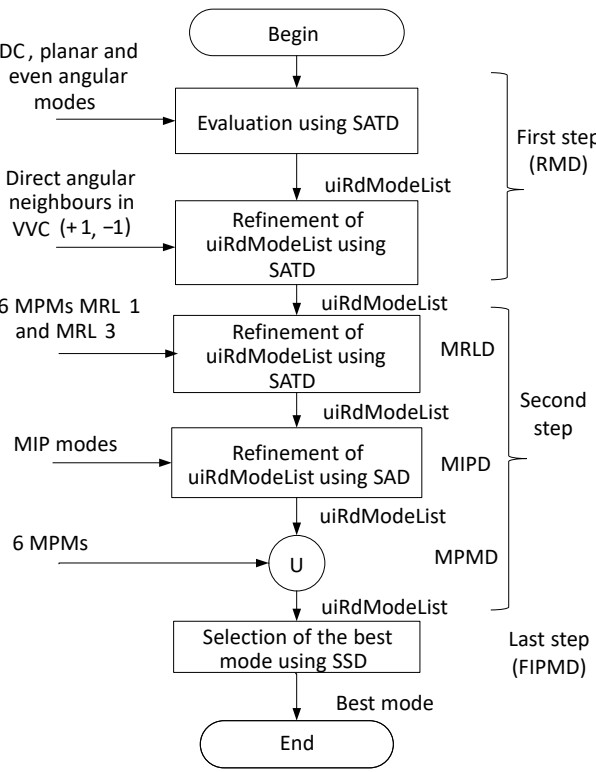

**Figure 3.** VVC intra-mode decision; uiRdModeList is the list of *N* candidates; 6 MPMs MRL 1 and 3 are the lists of 6 MPMs of farther reference lines 1 and 3, respectively.

First, the Rough Mode Decision (RMD) is applied to the 67 regular IPMs to select a set of *N* candidates (uiRdModeList), where *N* depends on whether the MIP is enabled for the current block or not. In essence, RMD usually selects 2 to 3 candidates, but when the MIP is used, it can refine the number of candidates by adding $log_2(min(H, W)) - 1$, where *H* and *W* are the height and width of the current block, respectively. To get the *J* cost, the Sum of Absolute Transform Difference (SATD), given by Equation (2) and the Truncated Binary Coding (TBC) [5] are used as the distortion and bit rate metrics, respectively.

$$SATD = \sum_{i,j} |A(i,j)|. \tag{2}$$

The matrix *A* is defined by Equation (3). It represents a matrix multiplication of the residual with the Hadamard transform *H*,

$$A = H \cdot (Y - \hat{Y}) \cdot H^T, \tag{3}$$

where, *Y* is the original block, $\hat{Y}$ is the intra-predicted block, and $H^T$ refers to the matrix transpose of *H*.

Secondly, the intra-coding tools have undergone RDO to update the set of *N* candidates by using three main steps, namely Multi-Reference Lines Decision (MRLD), Matrix weighted Intra Prediction Decision (MIPD), and Most Probable Modes Decision (MPMD). For instance, the MPMs are of 6 variable length IPMs derived according to the IPMs of left and above neighbouring blocks [5]. Hence, the MRLD tests the two lists of 6 MPMs: MRL1 and MRL3 using CABAC to infer the number of required bits *R* [5], and SATD as the distortion metric *D*. Then, the MIPD carries out the RDO on MIP modes with TBC and the Sum of Absolute Difference (SAD), given by Equation (4). The last step, so called, MPMD adds the 6 regular MPMs to the *N* candidates if they are not already included [5].

$$SAD = \sum_{i,j} |Y(i,j) - \hat{Y}(i,j)| \tag{4}$$

Finally, the Final Intra Prediction Mode Decision (FIPMD) tests the *N* best candidates along with the ISP candidates to infer the best IPM. It uses the Sum of Squared Difference (SSD), calculated by Equation (6), as the distortion metric *D* and CABAC to get the required number of bits *R*. Unlike the aforementioned steps of the intra-mode decision, the cost *J* here is computed by using the reconstructed block instead of the intra-predicted block. Consequently, $\hat{Y}$ in Equation (5) is obtained following an overall encoding process. Additionally, bit rate is no longer determined from an entropy coding of the selected IPM but an entropy coding of the residual block $(Y - \hat{Y})$.

$$SSD = \sum_{i,j} |Y(i,j) - \hat{Y}(i,j)|^2 \tag{5}$$

## 3. Related Works

Since the H.264/Advanced Video Coding (AVC), RDO [10] has been a core building block of video encoders. It managed to provide significant coding capabilities through minimization of the cost *J*, expressed by Equation (1). Considering that the addition of new coding tools would necessarily incorporate new decisions to be made, the RDO should have become much slower and more complicated. Consequently, several works have been proposed to reduce the RDO overhead. Some were relying on video texture characteristics or Statistical Learning (SL) to decrease the decision set, whereas others have taken advantage of the recent advances on Machine Learning (ML) to terminate the decision process early. This section presents complexity reduction techniques from the state of the art focusing on the intra-mode decision and the Coding Tree Unit (CTU) partitioning. The reviewed works are summarized in Table 1.

In [11], Y. Chen et al. investigated several correlations in the intra-mode decision of VVC. Consequently, two strategies were proposed in [11]. The first strategy uses the correlation between the 6 MPMs and the best IPM in RMD in order to reduce the decision set. For the second strategy, early termination of the intra-mode decision is evoked based on the difference between the best cost in RMD and the cost of the best IPM. Simulations show that the combination of these two strategies can reduce, on average, 30.59% of computational complexity with a slight increase in bit rate. J. Park et al. [6] made full use of block shape to infer a preprunable range of IPMs, that can be skipped during the test of ISP. Experimental results show that this method can save up to 12% of encoding time with almost no effect on the coding efficiency. The prediction distortion also plays a major role in reducing the RDO overhead. It has been exploited in [12] to terminate the Multi-type Tree (MT) partitioning early while providing a tunable decision framework. Simulations showed that the speed up may vary between 22.6% and 67.6% with birate loss ranging from 0.56% to 2.61%. Texture characteristics were also as useful as correlations in video content. It has been explored in many ways in order to accelerate the encoding decisions. For example, in [13] the number of candidates for the QTMT structure and the intra-mode decision are reduced according to texture characteristics. Basically, gradient difference is used to skip unlikely splits. However, the number of IPMs is reduced according to Coding Unit (CU) Texture Complexity (TC). This method can offer nearly 49% of speedup with a negligible loss on coding efficiency. Lui et al. [7] also proposed a fast-decision algorithm for the ISP by using the CU TC. They first classify the CUs into homogeneous and textured categories. Then, they terminate the ISP tests for homogeneous CUs. This solution can accelerate RDO by 7% with minimal loss on coding efficiency.

As for ML algorithms, they were able to achieve a significant tradeoff between computational complexity and coding efficiency, especially when trained on massive video data. In [14], for example, a ML approach based on Random Forest Classifier (RFC) was used to reduce the computational complexity of the intra-mode decision for both HEVC and VVC. In essence, Ryu et al. replace the best IPM in RMD with the IPM predicted by the RFC. Experimental results show that this algorithm allows 18% to 30% of complexity reduction at the cost of 0.70% of bit rate loss. Another fast-decision algorithm based on RFC was designed in [15]. It includes the RFC-based CU size decision and texture-based intra-mode

decision. The first algorithm relies essentially on the CU TC to decide whether to split the current CU or not. Then, the IPMs to be tested are selected according to the texture direction of the CU. These two algorithms offered nearly 54.91% of complexity reduction. As RFC presents a combination of several dependent Decision Trees (DT), it reveals that DT could also be convenient in speeding up the encoding process. Consequently, H. Yang, et al. proposed in [16] two main strategies to select the CU size and the best IPM. The first strategy consists of ignoring some splitting modes based on the DT predictions. The second approach is introduced to get the best IPM. This latter uses the well-known Gradient Descent Search (GDS). Hence, if all neighbouring blocks of the current CU exist, the GDS is performed to find the best IPM.

**Table 1.** Previous proposals.

| Proposal | SL | ML/DL | Intra | Partitioning | Achieved Results $\Delta T$ (%) | BD-BR (%) |
|:---:|:---:|:---:|:---:|:---:|:---:|:---:|
| [11] | × | | × | | 30.59 | 0.86 |
| [6] | × | | × | | 12.00 | 0.40 |
| [12] | × | | | × | 22.60–67.60 | 0.56–2.61 |
| [13] | × | | × | × | 46.00 | 0.91 |
| [7] | × | | × | | 7.00 | 0.09 |
| [14] | | × | × | | 18.00–30.00 | 0.70 |
| [15] | × | × | × | × | 54.91 | 0.93 |
| [16] | × | × | × | × | 70.00 | 1.93 |
| [17] | | × | | × | 46.60–69.80 | 0.86–2.57 |

Experimental results show that these two proposals can accomplish about 70% of time saving while slightly affecting the coding efficiency. Previous work [17] introduces a two-stage learning-based QTMT framework. This latter includes a CNN to predict the spatial features of an entire $64 \times 64$ luma block and a DT to infer the most likely splits at each sub-block. Simulations show that this framework can achieve up to 69.8% of complexity reduction with a small decrease in coding efficiency.

To conclude, all the previous proposals proved their efficiency in alleviating the complexity overhead of the VVC. However, our method has more advantages. First, it deals effectively with the high diversity of block shapes in the QTMT structure through predicting the probability at each $4 \times 4$ sub-block. Second, it opens up hardware optimization opportunities, like parallelism and GPU acceleration, by carrying out the prediction once for an entire $64 \times 64$ CU. Finally, it reduces the model inferring time by simultaneously predicting the probability vectors of all the intra-coding tools.

## 4. Multitask Learning-Based Intra-Mode Decision Framework

As detailed in Section 2, VVC proposes several intra-coding tools to enhance the intra-prediction. These tools have undergone RDO in order to decide the best coding tool and the Intra Prediction Mode (IPM) for each CU. The proposed method deals with deciding the best IPM during the FIPMD, which represents the most time consuming part of the intra-mode decision [11]. It introduces a MTL CNN to skip unlikely IPMs based on the inferred top-2 intra-coding tools.

### 4.1. Overall Presentation of the Proposed Framework

Multi-task Learning (MTL) has recently made it possible to predict several related tasks simultaneously by using a shared model, that leverages common acknowledge among all tasks [18].

As shown in Figure 4, our proposed method takes advantage of this learning paradigm to reduce the number of candidates for the FIPMD according to the inferred top-2 intra-coding tools. It feeds the original luma block, denoted as *B*, to MTL CNN in order to simultaneously predict whether to skip Matrix weighted Intra Prediction (MIP), regular, Multi-Reference Lines (MRL) and/or planar, DC. To accommodate the high diversity of block shapes in the QTMT partitioning structure, the CNN processes an entire $64 \times 64$ CU,

padded with the four MRL references lines, to output a probability vector $\hat{p}_t$ of usage at the $4 \times 4$ sub-blocks, for each intra-coding tool $t \in \{\text{MIP, regular, MRL, DC, planar}\}$. We have

$$\hat{p}_t = f_\theta(B), \qquad (6)$$

where $f_\theta$ is a parametric function, with learnable parameters $\theta$, used to approximate the output vector $\hat{p}_t$ for each coding tool $t$. The length of the output vector $\hat{p}_t$ is defined as

$$L = \left(\frac{S_B}{4}\right)^2, \qquad (7)$$

where $S$ is the block size. Hence, for a $64 \times 64$ luma block $L_p$ would be equal to 256. Figure 5 explains the representation of output vectors $\hat{p}_t$.

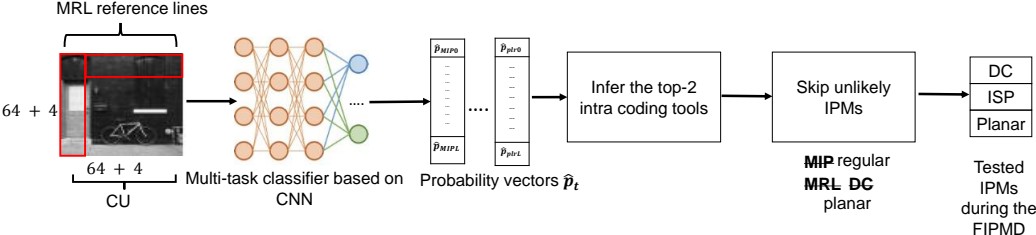

**Figure 4.** Workflow of MTL-based intra-mode decision framework. First, the luma block is fed to the MTL CNN to predict the probability vectors $\hat{p}_{MIP}$, $\hat{p}_{Reg}$, $\hat{p}_{MRL}$, $\hat{p}_{DC}$ and $\hat{p}_{Plr}$. Then, the top-2 most likely intra-coding tools are inferred with the maximums of mean probabilities at each block. Finally, the FIPMD is performed only on IPMs of the top-2 intra-coding tools in order to decide the best IPM.

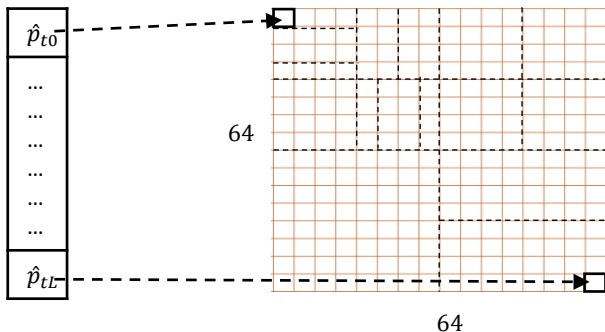

**Figure 5.** Representation of MTL CNN output vector, $L$ is length of probability vector $\hat{p}_t$, and t is the intra-coding tool.

As shown in this figure, $\hat{p}_{t0}$ is the probability of using an intra-coding tool $t$ at the first $4 \times 4$ sub-block of the $64 \times 64$ luma block and $\hat{p}_{tL}$ is the probability for the last $4 \times 4$ sub-block. These probability vectors are then used to skip unlikely IPMs as follows. First the probability of using MIP, regular IPMs, MRL, DC, and planar features are computed as the mean probabilities at each CU, denoted as $\mu_{mip}$, $\mu_{Reg}$, $\mu_{mrl}$, $\mu_{DC}$, and $\mu_{plr}$, respectively. Then, these latter are used to infer the top-2 intra-coding tools in each CU. As defined in Equation (8), the top-1 intra-coding tool is inferred as the maximum of mean probabilities in each CU. Then, a second-best intra-coding tool is deduced also as the maximum of mean probabilities, when excluding the top-1 intra-coding tool. We have

$$p_{top1} = max(\mu_{mip}, \mu_{Reg}, \mu_{mrl}, \mu_{DC}, \mu_{Plr}). \qquad (8)$$

Finally, the FIPMD is performed only on IPMs of the inferred top-2 intra-coding tools in order to decide the best IPM. For instance, the model was integrated under the VTM10.2 throughout the process illustrated in Figure 6.

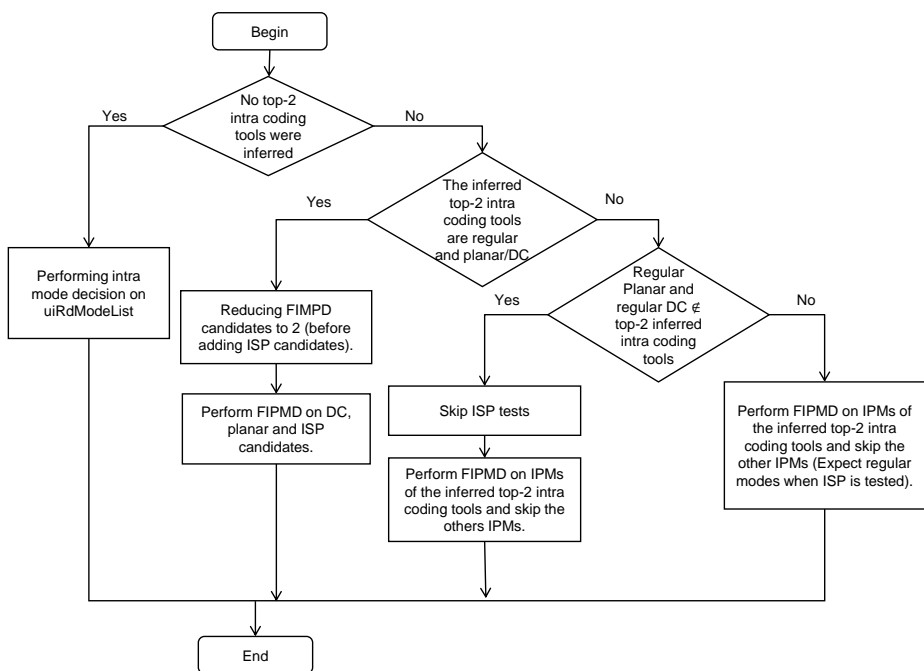

**Figure 6.** Process of integrating the MTL CNN under VTM10.2 to skip unlikely IPMs.

First, the model prediction is considered only when two top intra-coding tools are inferred at the current CU. Therefore, if these tools are regular IPMs and DC or planar, the number of tested IPMs for the FIPMD is reduced to 2. Therefore, only DC and planar with the ISP candidates are tested. Secondly, if DC and planar are not among the inferred top-2 intra-coding tools, then ISP tests can be skipped. In fact, the ISP usually reproduce 2 or 4 homogeneous regions, which will probably be coded with nonangular modes as demonstrated in [19]. Finally, the IPMs other than the top-2 inferred intra-tools are skipped. It worth mentioning that the MRL was excluded from the model prediction, because it does not allow a significant complexity reduction [20].

*4.2. Dataset and Training Process*

In this section, our dataset representations are given. Then, the proposed MTL CNN with its training process are detailed.

4.2.1. Training Dataset

Due to the lack of public datasets for our classification tasks, a dataset that yields encoded CUs with the best intra-mode information is established. Because the proposed method focuses on AI configuration, the image public dataset Div2k [21] was selected to derive the training samples.

In order to be encoded by using VTM [22], the images of this latter were concatenated as a pseudo video. Hence, they were encoded under the AI configuration for 4 Quantization Parameter (QP) (e.g., 22, 27, 32 and 37). Considering that the VTM includes several fast-decision techniques for the partitioning [23] and the intra-coding tools as well [24,25], these latter were disabled in order to build our training dataset. Disabling these techniques would ensure sufficiently diverse data for more accurate predictions.

The best intra-modes were first extracted from VTM encoder as trees, where each leaf node has its best intra-mode information, including the used intra-coding tool and IPM. Then, these latter were converted, as illustrated in Figure 5, into 256 probability vectors $\hat{p}_t$, that depict whether MIP, MRL, regular, and/or DC or planar are used for each $4 \times 4$ sub-block in the $64 \times 64$ luma block. Our dataset has 1.2 million samples distributed as given in Figure 7. As shown in this figure, the regular IPMs represent the majority of

training samples, with at least 77%, whereas MRL samples are under-represented, which is due to the fact that the VTM encoder tends to usually select regular IPMs [19].

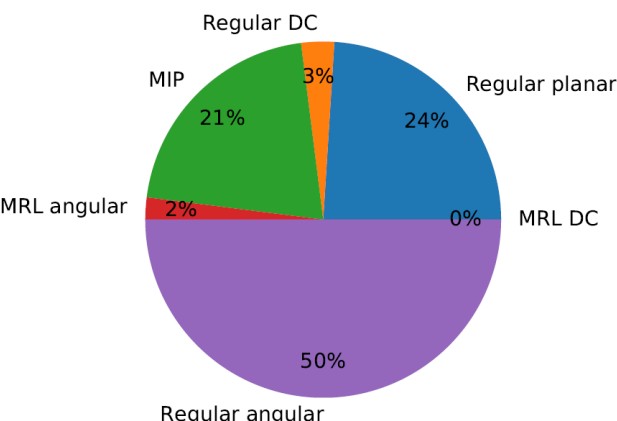

**Figure 7.** Distribution of training dataset.

4.2.2. Training Process

The MTL CNN, as illustrated in Figure 8 was inspired by the well-known ResNet [26].

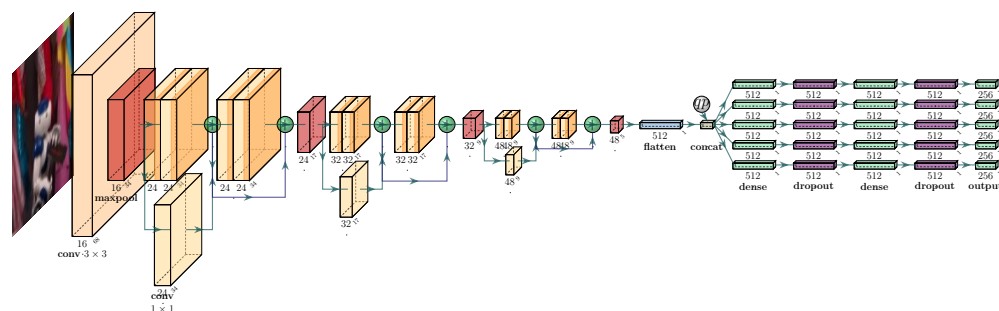

**Figure 8.** The MTL CNN architecture. Convolution layers are in orange and yellow, max-pooling layers in red, dropout in purple, and fully connected layers in cyan.

In addition, hard sharing of feature extraction layers [27] is adopted to ensure feature space sharing and performance across all tasks. The network was trained from scratch on the aforementioned dataset by using the Adam optimizer [28] to learn the optimum weights $\theta$ and minimize the sum of loss functions $l_t$ defined in Equation (9),

$$l_t = p_t(\log(\hat{p}_t) + \alpha(1 - p_t)(-\log(1 - \hat{p}_t)), \qquad (9)$$

where $t$ is the intra-coding tool and $\alpha$ is the weight assigned to the positive class. This latter is equal to: 3.5 for MIP, 1 for MRL and regular, 11.5 for DC, and 4.5 for planar.

This loss function was chosen due the highly skewed dataset we are dealing with, as explained in Section 4.2.1. Early stopping regularization was also used to ensure model convergence. Thus, our MTL CNN was trained for 14 epochs with a batch size of 128 and a learning rate of $10^{-3}$.

*4.3. Experimental Setup*

To assess the performance of our method, simulations are conducted on the reference software VTM10.2 [22] with the Common Test Conditions (CTC) [29] video sequences. Indeed, a set of 26 videos from classes A1, A2, B, C, D, E, and F are encoded under the AI configuration for 4 QP, which are 22, 27, 32 and 37. The number of encoded frames in each video is set to one Group of Pictures (GOP), which is equivalent to one second.

The effectiveness of our method is assessed by using both complexity reduction and coding efficiency. Hence, the Bjontegaard bit rate (BD-BR) [30] is used to measure the coding efficiency; then the average of run-time difference $\Delta T$, defined by Equation (10) is used to give the achieved complexity reduction. We have

$$\Delta T = \frac{1}{4} \sum_{QP_i=22,27,32,37} \frac{T_o(QP_i) - T_r(QP_i)}{T_o}, \tag{10}$$

where $T_o$ and $T_r$ represent the total encoder runtimes of the VTM anchor and the VTM when using the MTL CNN, respectively.

The MTL CNN was built and trained under the Keras framework, then converted with the frugally deep library [31] to C++ code, in order to be used in the VTM encoder.

### 4.3.1. Performance of the MTL CNN

Because our dataset is highly skewed, our model performance was evaluated with recall, precision, and F1 score in order to get a more informative picture of its performance, instead of using accuracy and loss metrics.

First, our model was evaluated under the default classification threshold, which is equal to 0.5. Then recall-precision curves, shown in Figure 9, were used to infer the best classification thresholds, which maximize the F1 score. For instance, the precision vs. recall curves of our model are illustrated as orange curves. The dashed blue curves give the precision vs. the recall of a no-skill model that outputs random guesses or predictions. Hence, the higher is the area under the curve of precision vs. recall, the higher is the tradeoff between precision and recall (e.g., F1 score).

Table 2 reports the aforementioned metrics for the default threshold and the best thresholds, which were inferred as 0.23 for MIP, 0.41 for regular, 0.13 for MRL, 0.46 for DC, and 0.42 for planar. As shown in this table, regular IPMs present the highest F1 score, which indicates that the model can generalize well in this classification task. For instance, the regular IPMs are highly presented in the training dataset, which is due to the encoder behavior. Indeed, the encoder tends usually to select the regular IPM, and this is clearly illustrated in Figure 7. However, for the remaining intra-coding tools, the MTL CNN prediction is quietly closer to the random guess or the no skill predictions, which is due to remaining tools being underrepresented in the training dataset. As mentioned, the encoder relies essentially on the 67 regular IPMs to perform the intra-prediction. Then, additional intra-coding tools may be considered to further enhance the coding efficiency, which is not always the case, as explained in [19]. For instance, under the default threshold the model has a low precision for DC, planar with a good recall. However, for MRL, it tends to have good precision and very low recall. This latter can be improved by varying the classification thresholds. However, the precision would become lower, which may introduce some loss in the coding efficiency. For this reason, the flowchart introduced in Figure 6 has been adopted to integrate the model under the VTM10.2 with minimum coding loss.

**Table 2.** Performance of the MTL CNN.

| Coding Tool | Default Threshold | | | Best Threshold | | |
|---|---|---|---|---|---|---|
| | Precision | Recall | F1-Score | Precision | Recall | F1-Score |
| Regular | 0.72 | 0.97 | 0.82 | 0.71 | 0.99 | 0.83 |
| MIP | 0.41 | 0.56 | 0.48 | 0.38 | 0.67 | 0.49 |
| MRL | 0.64 | 0.00 | 0.01 | 0.22 | 0.39 | 0.28 |
| DC | 0.19 | 0.39 | 0.26 | 0.19 | 0.40 | 0.26 |
| planar | 0.35 | 0.61 | 0.45 | 0.33 | 0.73 | 0.45 |

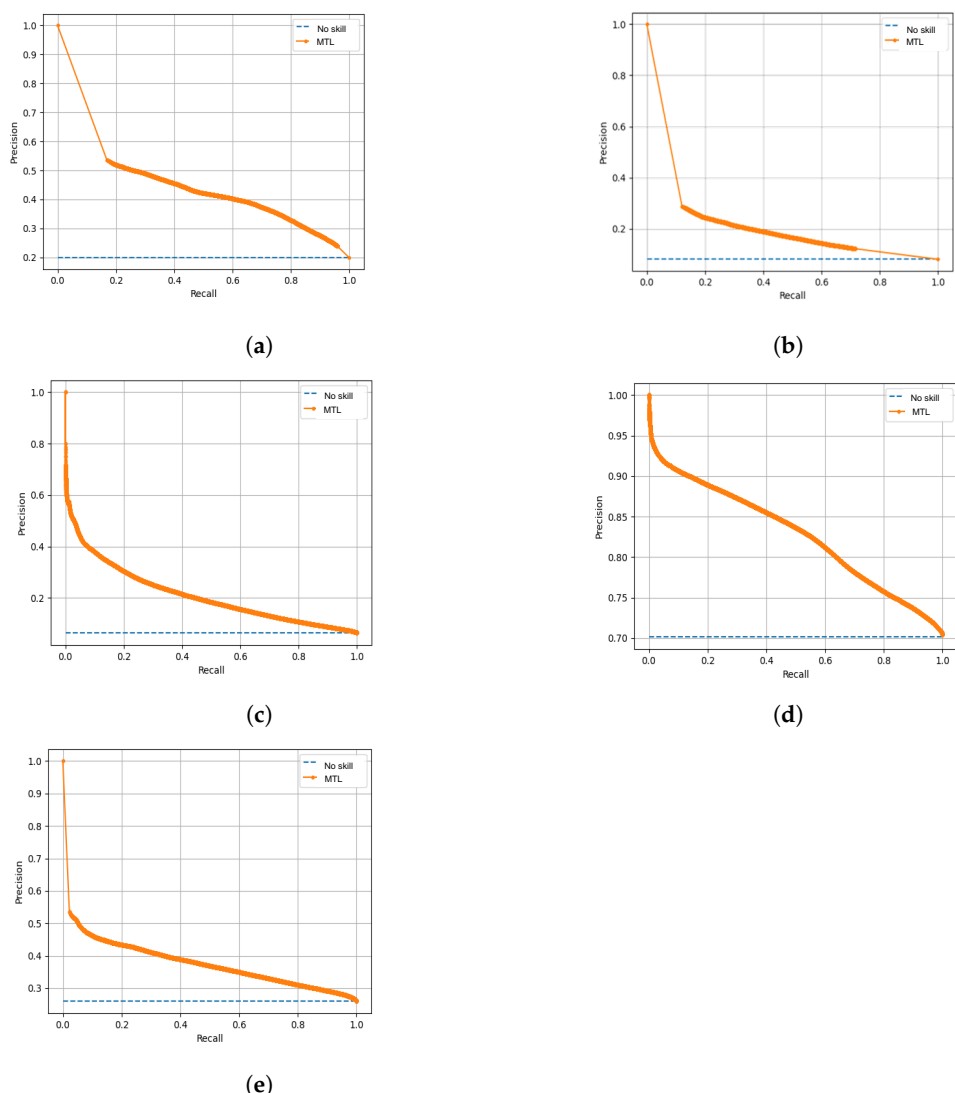

**Figure 9.** Precision vs recall curves: (**a**) for MIP; (**b**) for DC; (**c**) for MRL; (**d**) for regular; (**e**) for planar.

### 4.3.2. Complexity Reduction under VTM

Table 3 gives the performance of our MTL-based intra-mode decision framework in comparison with the state-of-the-art techniques. Indeed, all the native speed-up techniques included in VTM10.2 are activated to ensure fair comparison. As reported in Table 3, our MTL intra-mode decision framework can achieve on average 25.21% of complexity reduction for only 1.33% of BD-BR increase. Compared to the state of the art, our method outperforms [6,7] with about 13% and 19% of complexity reduction on average for a BD-BR increase of 0.9% and 1.02, which is considered as tolerated because our method can achieve 2 to 3 times more speedup. The method closest in performance to our own is [11], which enables 30.10% of complexity reduction on average. Indeed, this method picks in advance the best IPM and it is implemented under the VTM2.0, This is an old version of the VTM, which does not use either MIP, MRL, or ISP. However, our method is able to achieve closer performance for a more complex encoder when only predicting the set of IPMs to be tested during the FIPMD. For instance, the use of MIP and ISP enhanced the coding efficiency with about 2% [32] at the expense of 10% and 15% [33] of complexity overhead, respectively. The best performance of our method is presented for the video RaceHorsesC with about 30% of complexity reduction for only 1.03% of BD-BR increase. However, the worst case is presented for the video SlideEditting, which is a screen content. Indeed, the performance of our method, especially in term of BD-BR, decreases with the video resolution and this is directly related to the training dataset. In essence, the model is trained only with 4K fixed

images, which reduce its performance for lower video resolution. Moreover, no screen contents were used to train the model, which makes it not optimized for this type of content. However, our method allows a good trade off between complexity reduction and coding efficiency. Hence, varying video resolution and content may improve further our model performance under the VTM10.2.

**Table 3.** Performance of the MT based intra-mode decision framework in comparison with the state-of-the-art techniques.

| Class | Video | Chen, Y. et al. [11], VTM2.0 | | Park, J. et al. [6], VTM9.0 | | Li, Y. et al. [7], VTM8.0 | | Our Proposal, VTM10.2 | |
|---|---|---|---|---|---|---|---|---|---|
| | | $\Delta T$ (%) | BD-BR (%) | $\Delta T$ (%) | BD-BR (%) | $\Delta T$ (%) | BD-BR (%) | $\Delta T$ (%) | BD-BR (%) |
| A1 | Campfire | 28.06 | 0.92 | 12.00 | 0.09 | - | - | 24.54 | 0.78 |
| | Tango2 | 23.39 | 0.93 | 11.00 | 0.09 | - | - | 23.13 | 0.98 |
| | FoodMarket4 | 20.13 | 0.64 | 10.00 | 0.09 | - | - | 22.43 | 0.91 |
| | **Average** | **23.86** | **0.83** | **11.00** | **0.09** | **-** | **-** | **23.37** | **0.89** |
| A2 | CatRobot1 | 26.89 | 0.94 | 12.00 | 0.30 | - | - | 23.43 | 1.13 |
| | DaylighRoad2 | 32.99 | 0.98 | 11.00 | 0.49 | - | - | 24.64 | 1.59 |
| | ParkRunning3 | 20.32 | 0.67 | 9.00 | 0.07 | - | - | 20.63 | 0.59 |
| | **Average** | **26.73** | **0.86** | **10.67** | **0.29** | **-** | **-** | **22.89** | **1.10** |
| B | MarketPlace | - | - | 12.00 | 0.13 | - | - | 25.99 | 1.02 |
| | RitualDance | - | - | 12.00 | 0.32 | - | - | 22.98 | 1.25 |
| | Cactus | 29.47 | 0.54 | 13.00 | 0.49 | 6.00 | 0.14 | 27.11 | 1.36 |
| | BasketBallDrive | 34.69 | 0.51 | 11.00 | 0.64 | 9.00 | 0.24 | 24.70 | 1.82 |
| | BQTerrace | 37.17 | 0.44 | 12.00 | 0.48 | 4.00 | 0.01 | 26.94 | 0.49 |
| | **Average** | **33.78** | **0.50** | **12.00** | **0.41** | **6.33** | **0.13** | **25.54** | **1.19** |
| C | RaceHorses | 43.69 | 0.56 | 12.00 | 0.37 | 6.00 | 0.07 | 29.87 | 1.03 |
| | BasketBallDrill | 41.28 | 0.36 | 16.00 | 1.02 | 11.00 | 0.30 | 28.12 | 1.52 |
| | BQMall | 27.64 | 0.61 | 12.00 | 0.88 | 6.00 | 0.10 | 26.13 | 1.74 |
| | PartyScene | 43.69 | 0.56 | 13.00 | 0.49 | 4.00 | 0.01 | 27.97 | 1.24 |
| | **Average** | **39.34** | **0.50** | **13.25** | **0.69** | **6.75** | **0.48** | **28.02** | **1.38** |
| D | RaceHorses | 28.05 | 0.73 | 14.00 | 0.39 | 5.00 | 0.12 | 28.74 | 2.04 |
| | BQSquare | 30.08 | 0.61 | 12.00 | 0.57 | 8.00 | 0.18 | 27.82 | 1.45 |
| | BlowingBubbles | 29.09 | 0.70 | 14.00 | 0.65 | 6.00 | 0.00 | 26.53 | 1.56 |
| | BasketBallPass | 26.49 | 0.49 | 12.00 | 0.66 | 8.00 | 0.04 | 22.96 | 1.42 |
| | **Average** | **28.43** | **0.90** | **13.00** | **0.57** | **6.75** | **0.085** | **26.51** | **1.62** |
| E | FourPeople | 26.32 | 0.66 | - | - | 7.00 | 0.17 | 23.63 | 1.73 |
| | Johny | 25.85 | 0.59 | - | - | 8.00 | 0.22 | 22.95 | 1.72 |
| | KristenAndSara | 26.77 | 0.59 | - | - | 8.00 | 0.10 | 23.50 | 1.95 |
| | **Average** | **26.31** | **0.61** | **-** | **-** | **7.67** | **0.16** | **23.36** | **1.80** |
| **Average** | | **30.10** | **0.65** | **12.10** | **0.43** | **6.86** | **0.12** | **25.21** | **1.33** |
| F | ArenaOfValor | - | - | - | - | - | - | 24.09 | 1.61 |
| | BasketBallDrillText | 24.96 | 0.44 | - | - | - | - | 24.24 | 1.66 |
| | SlideEditting | 32.33 | 0.84 | - | - | - | - | 17.67 | 1.89 |
| | SlideShow | 32.50 | 0.66 | - | - | - | - | 20.78 | 1.92 |
| | **Average** | **29.93** | **0.65** | **-** | **-** | **-** | **-** | **21.69** | **1.77** |

## 5. Conclusions

In this paper, a MTL-based intra-mode decision framework for VVC is proposed. It deals with deciding the best IPM by reducing the set of FIPMD candidates according to the top-2 inferred intra-coding tools. Simulations proved that our method can achieve up to 30% of complexity reduction with a slight increase of BD-BR. As future work, we will consider improving further our method performance for low video resolutions and screen content. In addition, we will develop a complexity-reduction technique for Random Access (RA) configuration and interprediction.

**Author Contributions:** N.Z. designed, coordinated this research, drafted the manuscript and conducted the experiments and data analysis; D.M. and W.H. participated in the conceptualization, methodology, writing, revision, editing, assisted in data analysis and participated in the coordination of the research; A.K. and N.M. assisted in the data analysis, participated in the coordination of the research, supervision, writing, revision and editing. All authors have read and agreed to the published version of the manuscript.

**Funding:** This study was funded by France Campus with the PHC Maghreb project Eco-VVC (Grant number: 45988WG).

**Data Availability Statement:** The data that support the findings of this study are available from the corresponding author Zouidi, N upon reasonable request.

**Acknowledgments:** This work is supported by the France Campus, and within a co-supervised thesis between the Institute of Electronics and Telecommunications of Rennes, france (IETR), and the Laboratory of Electronics and Information Technology (LETI) of Sfax, Tunisia.

**Conflicts of Interest:** N.Z.; A.K.; W.H.; N.M. and D.M. declare that they have no conflict of interest.

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
