# Peer review of "Multitask Learning Based Intra-Mode Decision Framework for Versatile Video Coding"

_electronics, doi:10.3390/electronics11234001_

Round 1
Reviewer 1 Report
The paper deals with reducing the computational complexity of the VVC compression standard. It established a large database for intra prediction and proposes a Multi-task Learning based intra mode decision framework. Experimental results showed that the proposal enables up to 30% of complexity reduction while slightly increasing the Bjontegaard Bitrate (BD-BR).
The whole paper is logically divided into five chapters including the introduction. The paper is written adequate to the research domain and sound good from the technical point of view.
Considering my positive statements, I fully recommend to accept the paper in present mode.
Author Response
Thank you so much for your recommendation.
Reviewer 2 Report
The manuscript presents the use of CNN to estimate probabilities of intra-prediction modes. Results are applied to VTM of VVC to limit its computational complexity. Results show that the reduction is above 21%, and compression efficiency loss is 1.66% on average. This proves the efficiency and the membership to the set of the best reduction methods. The manuscript is well-written and clear.
Please explain the descriptions „no skill” and “logistic” in Fig. 9.
Author Response
Thank you for your comment
Point 1: the article was corrected accordingly and an additional proof reading has been performed.
Point 2: logistic was replaced by MTL to refer to the precision vs recall curves of our model and no skill model was explained as follow:
a no skill model is a model that outputs random guesses or predictions.